# Do Papers with Japanese Authors Have a Different Number of Authors? A Follow-Up Study after 25 Years and Implication for Other Countries

Akira Akabayashi [1,2,*], Eisuke Nakazawa [1] and Katsumi Mori [1]

1 Department of Biomedical Ethics, University of Tokyo Faculty of Medicine, 7-3-1 Hongo, Bunkyo-ku, Tokyo 113-0033, Japan
2 Division of Medical Ethics, New York University School of Medicine, 227 East 30th Street, New York, NY 10016, USA
* Correspondence: akirasan-tky@umin.ac.jp or akira.akabayashi@gmail.com

**Abstract:** A follow-up study was conducted 25 years after the 1997 *British Medical Journal* report. Articles with at least one Japanese author were defined as 'Article by Japanese' and those with no Japanese authors were defined as 'Article by Non-Japanese'. The number of authors per article for the years 2000, 2010, and 2020 in *Circulation*, *Circulation Research*, and the *Japanese Circulation Journal* was studied. Results are: (1) In all journals and all years covered, 'Article by Japanese' had more authors per article than 'Article by Non-Japanese'. Twenty-five years later, the results were similar. (2) Comparison by year revealed that all journals showed increases with time in the number of authors per article. We have discussed the problem of the Science Council of Japan's statement, influence on practising physicians and sample providers, and influence on international collaborations. This 25-year follow-up study highlights once again the need for global discussions on the qualifications for authorship in research studies.

**Keywords:** authorship; follow-up; twenty-five years; research ethics; science council of Japan

## 1. Introduction and Method

We conducted a follow-up study 25 years after the 1997 *British Medical Journal* report [1]. Our methods were similar to those of the aforementioned report and assessed authorship in *Circulation* (IF 39.9, basic), *Circulation Research* (IF 17.4, clinical), and *Japanese Circulation Journal* (IF 3.4, general). Articles with at least one Japanese author were defined as 'Article by Japanese' (JP) and those with no Japanese authors were defined as 'Article by Non-Japanese' (NJP). We counted the number of authors per article for the years 2000, 2010, and 2020.

## 2. Results

In all journals and all years covered.

(1) In all journals and all years covered, JP had more authors per article than NJP (Table 1). Twenty-five years later, the results were similar. For *Circulation*, in particular, the number of JP authors in 2020 was 1.64 times that of NJP authors. (Two-way ANOVA found JP to have significantly more authors than NJP in all journals, except for *Circulation* 2000 and 2010. Circulation [$F(5, 1556) = 74.79$, $p < 0.001$], Circulation Research [$F(5, 792) = 44.68$, $p < 0.001$], Japanese Circulation Journal [$F(3, 763) = 34.74$, $p < 0.001$]).

(2) Comparison by year revealed that all journals showed increases with time in the number of authors per article.

**Table 1.** Comparison of number of authors per article by Japanese (JP) and non-Japanese (NJP) categories.

| | Circulation | | Circulation Research | | (Japanese) Circulation Journal | |
|---|---|---|---|---|---|---|
| | JP | NJP | JP | NJP | JP | NJP |
| **2000:** | | | | | | |
| No of authors | 680 | 4566 | 491 | 1279 | 881 | 2 |
| No of articles | 79 | 658 | 69 | 227 | 121 | 1 |
| Mean No of authers/articles (range) | 8.6 (3–23) | 6.9 (1–21) | 7.1 (2–13) | 5.6 (1–12) | 7.3 (1–15) | 2.0 (2–2) |
| **2010:** | | | | | | |
| No of authors | 855 | 4558 | 703 | 1827 | 1999 | 801 |
| No of articles | 69 | 527 | 64 | 255 | 259 | 119 |
| Mean No of authers/articles (range) | 12.4 (2–61) | 8.6 (1–50) | 11.0 (2–23) | 7.2 (1–23) | 7.7 (1–24) | 6.7 (1–20) |
| **2020:** | | | | | | |
| No of authors | 854 | 3899 | 149 | 1993 | 2492 | 400 |
| No of articles | 27 | 202 | 11 | 172 | 216 | 51 |
| Mean No of authers/articles (range) | 31.6 (7–106) | 19.3 (2–241) | 13.5 (9–23) | 11.6 (2–44) | 11.5 (1–53) | 7.8 (1–19) |

## 3. Discussion

We will make three points.

(1) **Japan's recent trend**

The increase in the number of authors per paper is a global trend; since many researchers contribute significantly to labour-intensive studies, in molecular biology and genomics, for example, this is understandable. However, this trend is particularly pronounced in Japan. Fetters and Elwyn [1] cite Befu [2], who noted, "The Japanese penchant for 'groupism' and limited individual funding probably led them to involve more people in research endeavours. Research groups in Japan possess a cohesive sense of unity and mutual reliance on the group and senior leaders." We feel that this was all too lenient. The International Committee of Medical Journal Editors (ICMJE) has attempted to eliminate inappropriate authorship with more stringent criteria (Table 2). However, Japan has not adjusted accordingly. Why?

**Table 2.** Position of ICMJE, Nature and Science Council of Japan.

**ICMJE's position (2022, p2).**
**Who Is an Author?**
The ICMJE recommends that authorship be based on the following four criteria:
1. substantive contributions to the conception or design of the work; or the acquisition, analysis, or interpretation of data for the work; AND
2. drafting the work or revising it critically for important intellectual content; AND
3. final approval of the version to be published; AND
*Agreement to be accountable for all aspects of the work in ensuring that questions related to the accuracy or integrity of any part of the work are appropriately investigated and resolved.*
*In addition to being accountable for the parts of the work he or she has done, an author should be able to identify what co-authors are responsible for. In addition, authors should have confidence in the integrity of the contributions of their co-authors)*
*Nature*'s position (2022)
1. each author is expected to have made substantial contributions to the conception or design of the work; or the acquisition, analysis, or interpretation of data; or the creation of new software used in the work; or have drafted the work or substantively revised it
2. AND to have approved the submitted version (and any substantially modified version that involves the author's contribution to the study);
AND to have agreed both to be personally accountable for the author's own contributions and to ensure that questions related to the AND to have agreed both to be personally accountable for the author's own contributions and to ensure that questions related to the accuracy or integrity of any part of the work, even ones in which the author was not personally involved, are appropriately investigated, resolved, and the resolution documented in the literature.

**Table 2.** *Cont.*

| Science Council of Japan's position (2015, p2). |
| --- |
| 1. Substantial contributions to the design/conception of the work, or the execution of surveys/experiments; or substantial contributions to the work, such as the acquisition and analysis of experimental/observational data, or theoretical interpretation and model construction |
| 2. Contributions to the completion of the manuscript, such as writing a draft of the manuscript or expressing opinions |
| 3. Approval of the final version of the manuscript, and being accountable for the contents of the manuscript.See also the Note. |

Note: However, since these requirements are subject to broad interpretation depending on the area of research, judgement should be based on the consensus of the authors. When there are several authors, it is desirable that the roles each author played regarding the manuscript be clearly stated.

One influential factor may be a statement made by the Science Council of Japan (SCJ), the only official representative body of researchers in Japan supported by the government. In 2015, it stated the following:

> *In Japan, it has been the practice in some fields to add authors who have merely provided research equipment and facilities, provided funding, given authority to the article, or taught, suggested or advised on well-known theories, even if they do not "fulfil all the above requirements" for authorship. The reason for this is that in Japan, Acknowledgement tends to be considered a formality. In the future, it will be necessary to recognise the significance of references in Acknowledgements, as in Europe and the USA, delineating between authors responsible for the research results and those listed in the Acknowledgements . . .*

> *We, the committee, have read the ICMJE's "Defining the Role of Authors and Contributors" (revised April 2010). Our statement refers to the above (ICMJE) and offers our own views. (Authors' translation, p.2)*

The committee noted above consisted of 18 members, which included 10 well-established university professors, one director on the Board of the Institute of Natural Science, and seven administrative officials. The Japanese are generally very compliant with political authorization, and Japanese researchers may have felt that authorization by the SCJ gave them the freedom to increase the number of authors.

However, $H_2O$ is $H_2O$ everywhere. Natural science is common throughout the world. It is obvious that the SCJ missed the common aspects of natural science. If Japan continues to assume this unique position on this matter, it could be rejected by the international research community.

Here we consider Japan as an example; analyses of other countries might reveal similar violations. We firmly believe that all researchers worldwide should abide by international standards.

(2) **Influence on Practicing Physicians and Sample Providers**

The positions of the ICMJE, *Nature*, and SCJ are shown in Table 2 [3–5]. Stringency of the standards of both the ICMJE and *Nature* is ensured by Item 4 of the ICMJE standard and Item 3 of the *Nature* standard. Granted, stronger restrictions may disincentivize some physicians to participate in research. For example, suppose a practicing physician (i.e. a GP in the UK) was asked by a university researcher to provide the long clinical course of a rare disease as well as blood, urine, skin biopsy tissue, or other residual samples from patients in long-time care in order to study very rare genetic conditions with serious mental and physical symptoms. The physician has to explain the objective of the research study and obtain the patients' written informed consent, which would take a long time for the physician and would have to be done very carefully. The physician should not lose the trust of the patient or destroy a good long-term patient-physician relationship. As a result of this physician's efforts, the university researcher's paper is finally published in the high impact journal and the physician's name is not included as an author of the paper published but is mentioned in one sentence in the Acknowledgement. How would the physician feel? How would this influence practicing physicians' willingness to participate in these studies? This

also raises the question of why an important sample provider should not be included as an author since the research study will not be possible without those samples in the first place.

(3)     **Influence on International Collaborations**

Another serious issue concerns the authorship of international collaborative research studies. Considering the recent remarkable increase in international collaborations using human samples from multiple areas and countries, if the criteria of inclusion of the authorship differs, significant confusion will occur. Namely, the international collaborators who have contributed equally to a project, say in data curation, may or may not be included as authors. Inclusion as authors may be entirely dependent upon their country or area of origin. Especially in the case of international collaborative research, adherence to international standards is of the utmost importance.

## 4. Conclusions

Our results first of all suggest that international differences persist. This 25-year follow-up study highlights once again the need for global discussions on the qualifications for authorship in research studies.

**Author Contributions:** Conceptualization, A.A.; methodology, K.M. and E.N.; data curation: K.M. validation, E.N.; formal analysis, K.M; writing—original draft preparation, A.A.; writing—review and editing, E.N., K.M. and A.A. project administration, A.A.; All authors have read and agreed to the published version of the manuscript.

**Funding:** This research received no external funding.

**Data Availability Statement:** The data presented in this study are available on request from the corresponding author.

**Acknowledgments:** The authors thank Erika Shoji of the University of Tokyo for her help in writing this Communication.

**Conflicts of Interest:** The authors declare no conflict of interest.

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
