# Peer review of "Do Papers with Japanese Authors Have a Different Number of Authors? A Follow-Up Study after 25 Years and Implication for Other Countries"

_publications, doi:10.3390/publications10040038_

Round 1
Reviewer 1 Report
The base data of this short paper is moderately interesting, showing that the pattern of authorship in Japan differs from non-Japanese authors, and that this has not changed in 25 years The analysis and discussion is significantly lacking.
The emotive language is unhelpful…”Shockingly similar” lines 15 and 35. Japan is a “shallow” country line 47. Shamelessly line 99. This paper does not provide argument to support these emotive comments.
The central argument in this paper is that the attitude towards authorship in Japan is more permissive than that espoused by the ICMJE. They appropriately acknowledge the cultural norms in Japan that probably contribute to this, but do not discuss or acknowledge the cultural norms that contribute to the position of the ICMJE. Arguably the dominance of the US and its significantly individualistic culture contributes to narrowing of the criteria favouring individual researchers at the expense of collective groups; of trying to determine individual contribution compare to accepting that it was a team effort.
The BMJ paper that was cited was part of a much larger correspondence following an editorial[1] The editorial and the accompanying responses provided a nuanced and complex discussion of this issue that by no means asserted that any one approach was “right”. This paper refers to just one of the responses to this editorial and does not address the many points raised in both the editorial and the responses. There is little or no reference to any of the debate that has taken place in the intervening years.
The assertion that science is universal in line 114 is poorly argued by a single example (H20) A discussion of some of the papers in the journals that they reviewed might have been helpful, Papers documenting for example the effects of interventions (maybe the outcome of inserting stents) are likely to have a more universal applicability. Papers that involve value judgements and beliefs of patients or clinicians many not be.
They assert the importance or adhering to international standards but then discuss a circumstance (Lines 122-134) where the international standards are too stringent.
There is no significant discussion of the relative benefits and harms of having more or less authors, and in particular how that might affect Japanese researchers. The editorial which was the stimulus for the original paper that they were emulating was far from certain as to what the best approach might be.
There is an issue about international collaborations, but it is unclear the scale of that issue. It would seem likely that it will get resolved on a case by case basis. If it is a collaboration between Japanese and authors from another country and that Japanese authors propose twice as many authors for a collaborative paper than the other country, a negotiation between those authors will take place.
The analysis showing the difference between Japanese and non-Japanese authors is interesting and of value. If a non-Japanese institution was looking at the publication record of a Japanese applicant they would need to understand this difference for fear of placing an unreasonable weight on their publication statistics.
The comment in the final paragraph referring to Artificial Intelligence and authorship should either be deleted or expanded.This area of research is quite different from that publication in Circulation and its relevance to the data in this paper has not been discussed
1. Smith, R., Authorship: time for a paradigm shift? BMJ, 1997. 314(7086): p. 992.
Author Response
Rev 1 . Comments and Suggestions for Authors
The base data of this short paper is moderately interesting, showing that the pattern of authorship in Japan differs from non-Japanese authors, and that this has not changed in 25 years The analysis and discussion is significantly lacking.
The emotive language is unhelpful…”Shockingly similar” lines 15 and 35. Japan is a “shallow” country line 47. Shamelessly line 99. This paper does not provide argument to support these emotive comments.
Thank you for your useful comments. We agree, and have amended the emotive language.
”Shockingly similar” lines 15 and 35. Deleted.
Japan is a “shallow” country line 47. Changed to ‘Japan’s recent trend.’
“Shamelessly,” line 99. Deleted.
The central argument in this paper is that the attitude towards authorship in Japan is more permissive than that espoused by the ICMJE. They appropriately acknowledge the cultural norms in Japan that probably contribute to this, but do not discuss or acknowledge the cultural norms that contribute to the position of the ICMJE. Arguably the dominance of the US and its significantly individualistic culture contributes to narrowing of the criteria favouring individual researchers at the expense of collective groups; of trying to determine individual contribution compare to accepting that it was a team effort.
The BMJ paper that was cited was part of a much larger correspondence following an editorial[1] The editorial and the accompanying responses provided a nuanced and complex discussion of this issue that by no means asserted that any one approach was “right”. This paper refers to just one of the responses to this editorial and does not address the many points raised in both the editorial and the responses. There is little or no reference to any of the debate that has taken place in the intervening years.
Thank you for this comment. The editorial you mentioned was published in 1997, a quarter century ago. As you correctly say, there were many discussions at that time. However, in 2022, the situation has significantly changed from 1997, and many scientific journals require that the the ICMJE guidelines be followed, with some requiring ICMJE disclosure forms. In fact, your journal Publication is one of those journals. If researchers do not follow journal policy, the journal will not accept their papers. We believe the ICMJE guidelines have contributed to inappropriate authorship, such as gift authorship to some extent.
It seems you are criticizing the Western individualistic approach by ICMJE, and we also appreciate your tolerance towards Japanese collective society. But why can you say ‘They “appropriately” acknowledge the cultural norms in Japan?
The assertion that science is universal in line 114 is poorly argued by a single example (H20) A discussion of some of the papers in the journals that they reviewed might have been helpful, Papers documenting for example the effects of interventions (maybe the outcome of inserting stents) are likely to have a more universal applicability. Papers that involve value judgements and beliefs of patients or clinicians many not be.
We stated in the original manuscript, ‘Science, especially natural science, is universal.’ which was a bit misleading. We are assuming only natural science from the first. Therefore, we have changed the phrase as follows:
H2 O is H2 O everywhere. Natural science is common throughout the world.
They assert the importance or adhering to international standards but then discuss a circumstance (Lines 122-134) where the international standards are too stringent.
Of course, there are no perfect guidelines. This includes international standards such as ICMJE, and so ICMJE is updating. It is a trade-off. If the guidelines become strict, it may be an inhibiting factor. The ICMJE released a draft revised disclosure form in January 2020 and sought comments by the end of April of the same year. ICMJE is also asking for opinions from researchers.
There is no significant discussion of the relative benefits and harms of having more or less authors, and in particular how that might affect Japanese researchers. The editorial which was the stimulus for the original paper that they were emulating was far from certain as to what the best approach might be.
The relative benefits and harms of having more or fewer authors” is out of the scope of this short communication.
There is an issue about international collaborations, but it is unclear the scale of that issue. It would seem likely that it will get resolved on a case by case basis. If it is a collaboration between Japanese and authors from another country and that Japanese authors propose twice as many authors for a collaborative paper than the other country, a negotiation between those authors will take place.
There are many international collaborations, but the case-by-case negotiations, you mention seem to us a little optimistic. We have actually had challenging experiences in international collaborations.
The analysis showing the difference between Japanese and non-Japanese authors is interesting and of value. If a non-Japanese institution was looking at the publication record of a Japanese applicant they would need to understand this difference for fear of placing an unreasonable weight on their publication statistics.
Unfortunately, we are unable to reply to this comment.
The comment in the final paragraph referring to Artificial Intelligence and authorship should either be deleted or expanded. This area of research is quite different from that publication in Circulation and its relevance to the data in this paper has not been discussed
Yes, thank you for pointing this out. We have deleted it.
- Smith, R., Authorship: time for a paradigm shift?BMJ, 1997. 314(7086): p. 992.
We would like to emphasize that we are not at all universalist, and we are not simply supporting an individualistic society. We appreciate the collective society ethos where we live. However, at least in natural science, we believe we need some international standards for authorship. Otherwise, it is not fair, and there are significant risks in international collaboration. I hope you understand our position and that you feel comfortable with this revision for publication.
Once again, thank you for reading our paper.
Reviewer 2 Report
This an interesting study comparing mean number of authors per paper for Japanese (JP) and non-Japanese (NJP) authored papers for three time periods and three circulation journals. The data do support the conclusion that JP papers are likely to have more authors than non-JP papers. However, I have a few comments/concerns.
JP was defined as having at least one Japanese author while NJP was defined as have no Japanese authors. This definition needs further justification and discussion. For example, why would we expect one Japanese author to have an impact if there are 24 other authors? Why not define JP as a percentage of Japanese authors, e.g., 50% or more? Why not just not use the terms JP and NJP and quantify papers in terms of the percentage of Japanese authors? This could give a more precise test of the hypothesis.
Also, I saw no statistics for this paper. I would be helpful to do some statistical analysis to see how big the difference between JP and NJP is, whether there is a trend over time, and p-values for these statistics.
The title refers to culture but culture is a big, amorphous category that is not really being evaluated per se. I think what is being evaluated is whether papers have Japanese authors.
Author Response
Rev 2. Comments and Suggestions for Authors
This an interesting study comparing mean number of authors per paper for Japanese (JP) and non-Japanese (NJP) authored papers for three time periods and three circulation journals. The data do support the conclusion that JP papers are likely to have more authors than non-JP papers. However, I have a few comments/concerns.
JP was defined as having at least one Japanese author while NJP was defined as have no Japanese authors. This definition needs further justification and discussion. For example, why would we expect one Japanese author to have an impact if there are 24 other authors? Why not define JP as a percentage of Japanese authors, e.g., 50% or more? Why not just not use the terms JP and NJP and quantify papers in terms of the percentage of Japanese authors? This could give a more precise test of the hypothesis.
Thank you very much for your comment. The reasons we selected this categorization are two-fold.
1) To define JP as a percentage of Japanese authors is certainly one choice of definition. However, our statistic expert explained that it is very hard to determine the cut-off line. You say, e.g., 50%, but we need statistical justification to set the cut-off line at 50%. Otherwise, statistically, the -off line is criticized for being arbitrarily determined. If we set it at, say, 30% or 70%, the result could be totally different.
2) This is a follow-up study to the 1997 BMJ letter, so we used the same methods.
Because of these two reasons, we used the same method as the previous report.
Also, I saw no statistics for this paper. I would be helpful to do some statistical analysis to see how big the difference between JP and NJP is, whether there is a trend over time, and p-values for these statistics.
As written in the original manuscript result section, we conducted a two-way ANOVA. We have added F and p values in the result section. Circulation (F(5, 1556)=74.79, p<0.001)、Circulation Research (F(5, 792)=44.68, p<0.001), Japanese Circulation Journal (F(3, 763)=34.74, p<0.001). So, this data is statistically analyzed, and is more sophisticated than the 1997 BMJ letter.
The title refers to culture but culture is a big, amorphous category that is not really being evaluated per se. I think what is being evaluated is whether papers have Japanese authors.
Thank you for your comment. Since this is a follow-up of the 1997 BMJ letter, so we used the same title. But you are right. We have changed the title from:
Is authorship assessment dependent on culture? Follow-up after 25 years
To
Is the number of authorship different when Japanese authors are included? Follow-up study after 25 years and implication to other countries.
However, we are happy to change, and would welcome your suggestion for a new title
Once again, thank you for reading our paper.
Round 2
Reviewer 1 Report
Thanks for your responses I am happy to support publication
Author Response
Once again, thank you for reading our manuscript.
Reviewer 2 Report
The responses were good.
Author Response

(The authors gave the same response as above.)
